# Depression Treatment in Pregnancy: Is It Safe, or Is It Not?

**DOI:** 10.3390/ijerph21040404

**Published:** 2024-03-26

**Authors:** Vitalba Gallitelli, Rita Franco, Sofia Guidi, Ludovica Puri, Marco Parasiliti, Annalisa Vidiri, Georgios Eleftheriou, Federica Perelli, Anna Franca Cavaliere

**Affiliations:** 1Division of Gynecology and Obstetrics, Isola Tiberina Gemelli Hospital, 00186 Rome, Italy; vitalba.gallitelli.fw@fbf-isola.it (V.G.); ludovica.puri91@gmail.com (L.P.); mp.parasiliti@gmail.com (M.P.); annalisavidiri@gmail.com (A.V.); annafranca.cavaliere@fbf-isola.it (A.F.C.); 2Division of Gynecology and Obstetrics, IRCSS Azienda Ospedaliera-Universitaria of Bologna, 40138 Bologna, Italy; sofia.guidi4@studio.unibo.it; 3Poison Control Center, Hospital Papa Giovanni XXIII, 24127 Bergamo, Italy; jorgos_2002@yahoo.com; 4Azienda USL Toscana Centro, Gynecology and Obstetrics Department, Santa Maria Annunziata Hospital, 50012 Florence, Italy; federica.perelli@uslcentro.toscana.it

**Keywords:** pregnancy, depression, antidepressants, treatment, perinatal

## Abstract

Prenatal depression carries substantial risks for maternal and fetal health and increases susceptibility to postpartum depression. Untreated depression in pregnancy is correlated with adverse outcomes such as an increased risk of suicidal ideation, miscarriage and neonatal growth problems. Notwithstanding concerns about the use of antidepressants, the available treatment options emphasize the importance of specialized medical supervision during gestation. The purpose of this paper is to conduct a brief literature review on the main antidepressant drugs and their effects on pregnancy, assessing their risks and benefits. The analysis of the literature shows that it is essential that pregnancy be followed by specialized doctors and multidisciplinary teams (obstetricians, psychiatrists and psychologists) who attend to the woman’s needs. Depression can now be treated safely during pregnancy by choosing drugs that have no teratogenic effects and fewer side effects for both mother and child. Comprehensive strategies involving increased awareness, early diagnosis, clear guidelines and effective treatment are essential to mitigate the impact of perinatal depression.

## 1. Introduction

“No health without perinatal mental health” [1].

The perinatal period is a highly dynamic phase marked by alternating pleasant and worrisome thoughts and moods. Particularly in individuals deemed “fragile”, there is an increased possibility of developing or exacerbating mental disorders like depression, anxiety, post traumatic stress disorder and the more serious postpartum psychosis. Other psychiatric conditions more typical of the postpartum period include eating disorders (which may worsen, especially during weaning) and obsessive-compulsive disorder [2]. According to the World Health Organization, a mental disorder (mainly depression) may be faced by 10% of pregnant women and 13% of recently delivered women. In developing countries, depression rates are even higher, being 16% during pregnancy and 20% post-delivery. Prevalence studies on postpartum depression, while heterogeneous in design and methods, consistently reveal high global prevalence, estimated at 10–20% during the perinatal period [3,4,5]. However, it is crucial to note that due to women’s difficulty in recognizing and seeking help during the perinatal period, this problem may be underdiagnosed [6].

Depression is a disease that severely impacts how an individual functions. It is characterized, from a psychologic point of view, by irritability, sadness or empty moods and, from a somatic and cognitive point of view, by loss of concentration, anhedonia (lack of pleasure), hopelessness, sleep disturbances, loss of appetite and suicidal ideation. Depression occurring during pregnancy is termed prenatal or antenatal depression [7]. Prenatal depression warrants identification and treatment similar to postpartum depression.

Antenatal depression falls under the specifier “with peripartum onset”, with mood episodes that may include psychotic elements, and the criteria aimed at a diagnosis include a depressed mood, significantly reduced interest or pleasure in activities, feelings of worthlessness or inappropriate guilt, psychomotor retardation or agitation, impaired thinking or concentration and recurrent thoughts of death [8]. Somatic symptoms, such as appetite and weight changes, variations in sleep patterns and fatigue, may also occur in physiological pregnancies due to normal neuroendocrine changes [9].

Prenatal depression can have meaningful consequences for both the mother and the fetus. Its outcomes in a pregnancy are several, and they may include delayed fetal development, a low birth weight, spontaneous abortion, preterm labor or delivery and a cesarean section (CS). In addition, the outcomes on the mother’s health include anemia, diabetes, hypertensive disorders (including preeclampsia) and postpartum depression. For what concerns the neonatal outcomes, the neonates or infants born to mothers affected by depression during pregnancy tend to be more irritable, less active and prone to developmental delays. Furthermore, as previously mentioned, since prenatal depression is linked to an increased risk of developing postpartum depression, if the mother remains depressed for a protracted period of time without intervention, then this could potentially harm the infant’s health and growth. Notwithstanding the consequences of antenatal depression and the potential benefits of appropriate screening and treatment in reducing adverse fetal, maternal and neonatal outcomes, the vast majority of the scientific research has predominantly focused on postpartum depression [10].

Pregnancy significantly contributes to the discontinuation of antidepressant drugs. Women after six weeks of pregnancy do not receive additional antidepressant prescriptions. Depression screening in pregnancy is often ignored, despite the gravity of the disease, leading to a lack of necessary treatment and measures to prevent the worsening of symptoms. Frequently, when depression is diagnosed, it remains untreated [10]. Women should be screened and treated for depression before conception because otherwise, many may become pregnant with untreated depression.

While conflicting studies exist regarding the importance of obstetric complications in depressed women, it is widely acknowledged that untreated mental illnesses, including suicide attempts, contribute to maternal morbidity and mortality. Most researchers agree that untreated depression may be linked to obstetric complications and pathologies in the puerperium [11].

Untreated depression during pregnancy is linked to unfavorable outcomes like bleeding during gestation, spontaneous abortion, increased uterine artery resistance, low Apgar scores, neonatal care unit admission [12], neonatal growth retardation, spontaneous early labor, fetal death, a low birth weight, babies small for their gestational age, perinatal and birth complications, preterm deliveries and high cortisol levels in offspring at birth. Furthermore, subjective reports of labor being more painful, frequently requiring epidural analgesia and surgical deliveries such as cesarean sections or vaginal instrumental deliveries are associated with depression [13,14,15,16,17].

One of the most glaring concerns is the fact that untreated depression during pregnancy can aggravate the illness and increase the risk of suicidal thoughts and attempts.

If prenatal depression is left untreated, then there is a 50–62% chance of a post-partum episode and worsening of the mental disorder. Pregnancy termination on psychiatric grounds is not unusual [18,19].

Depression’s biological dysregulation may not be ideal for pregnancy. Nonetheless, rather than emphasizing the dangers of untreated depression, the majority of research focuses on the obstetric complications associated with the use of psychotropic medications. Few studies take nonpharmacologic therapy during pregnancy into account, leading women with depression to refuse treatment due to fears of teratogenic effects. Many are unaware that they may be putting their pregnancy and baby at risk, especially considering the limited availability of psychotherapy.

Apprehensions regarding the use of antidepressant drugs in pregnancy revolve around three main effects: (1) teratogenicity, (2) perinatal syndromes (neonatal toxicity) and (3) postnatal behavioral sequelae. While consistent data exist for many drug classes regarding the risk of teratogenicity [20], exposure in utero to certain psychotropic drugs may raise the risk of specific congenital anomalies. However, the occurrence rate of these congenital anomalies, even with an increased risk, remains low. The use of psychotropic drugs during pregnancy is justified in many clinical situations, and the risk of prenatal exposure should be carefully balanced against the risk of relapse after drug withdrawal.

Nevertheless, it is crucial that pregnancy is supervised by specialized doctors and multidisciplinary teams (obstetricians, psychiatrists and psychologists) addressing the woman’s needs. Depression can now be safely treated during pregnancy by selecting drugs without teratogenic effects and fewer side effects for both mother and child.

The aim of this paper is to conduct a brief literature narrative review on the main antidepressant drugs and their effects on pregnancy, assessing their risks and benefits. Below, we outline the primary classes of antidepressant drugs, describing their main effects on pregnancy and the fetus.

The characteristics of all the drug classes described are summarized in Table 1.

## 2. Drugs and Treatments

### 2.1. Tricyclic Antidepressants (TCAs): Clomipramine, Imipramine, Amitriptyline, Trimipramine, Desipramine, Nortriptyline, Perfenazine and Trazodone

Over 400 cases of first-trimester exposure to tricyclic antidepressants have been studied in 3 prospective and over 10 retrospective studies to determine the risk of organ malformation. These studies show no evidence of a significant correlation between fetal exposure to TCAs and the risk of any major congenital abnormality, either when analyzed on an individual basis and when pooled [21,22,23].

However, because of its undesirable side effects (sedation and anticholinergic side effects), this class of medicine is not recommended as a first line of treatment for mood disorders. Desipramine and nortriptyline are frequently chosen among the TCAs since they are less anti-cholinergic and the least likely to worsen orthostatic hypotension that occurs during pregnancy [24,25].

### 2.2. Serotonin Reuptake Inhibitors (SSRIs): Citalopram, Escitalopram, Fluoxetine, Paroxetine, Fluvoxamine and Sertraline

According to several teratology studies, the prevalence of malformations in offspring born to serotonin reuptake inhibitor-using mothers was similar to that of unexposed subjects [26,27].

The use of SSRIs during pregnancy has generated debate, since prior reports have linked exposure to these medications during the first trimester to an elevated risk of congenital heart defects and some subtypes of heart defects [28,29].

Several studies in the literature have demonstrated the reproductive safety of escitalopram, similar to the progenitor drug citalopram.

An increased risk of major malformations and heart defects has not been demonstrated, although the number of cases examined is not large enough to draw definitive conclusions [30,31,32].

Paroxetine use was linked to right ventricular outflow tract defects, while fluoxetine use was linked to isolated ventricular septal defects, but the absolute risks of both were small [33].

Other studies did not demonstrate the same increased risk of teratogenicity with exposure to fluoxetine and paroxetine in the first trimester.

Fluvoxamine and sertraline, new-generation SSRIs, could be used as second-choice drugs in some patients who have not responded to common antidepressants. Current data on these drugs are not as extensive. However, no congenital malformations have been shown in infants after exposure [26].

Therefore, although some continue to avoid these antidepressants as first-line drugs for a woman of reproductive age, given the encouraging evidence, these antidepressants should be taken into consideration as therapeutic options during pregnancy [34,35,36,37,38].

Among all SSRIs, escitalopram was associated with higher rates of low birth weight (<2500 g), although it is difficult to determine whether this effect was due to the depression itself or exposure to the drug [32].

In general, after adjustment for maternal depression, no correlation was observed between the use of serotonin reuptake inhibitors and preterm delivery or low birth weight [39,40].

Case reports, cohort studies, case-control studies and a data mining investigation of the WHO database of adverse drug reactions all described the adverse effects in neonates exposed to serotonin reuptake inhibitors during pregnancy [41,42,43,44,45,46]. Respiratory distress, jitteriness, irritability, vomiting, persistent pulmonary hypertension of the newborn and, in rare cases, convulsions were among the side effects. In most cases, the symptoms were minor and disappeared within two weeks of life with no treatment or with only supportive care.

The infant serotonin transporter promoter genotype may act as a mediator in the development of neonatal behavioral symptoms [47].

Though most of these studies did not use raters blinded to the mother’s treatment status, these studies deserve careful evaluation. A further limitation is the paucity of studies that evaluated the mother’s mood during gestation or at the moment of birth. It is crucial to assess how the maternal mood affects the newborn outcomes, because there is ample evidence indicating that anxiety or depression in the mother may be a factor in poor neonatal outcomes, including preterm birth and low birth weight. Based on these findings, many women are advised to taper or stop treatment with SSRIs before giving birth. However, this approach has not been shown to alter neonatal outcomes.

Notably, both treated and untreated mood and anxiety disorders have been linked to neonatal effects, and limited studies have adequately teased out these factors.

One crucial consideration is that the risk of postpartum depression may increase if medication is stopped or dosed down in the later stages of pregnancy.

An additional concern has been that maternal SSRI usage may be associated with a higher-than-expected number of cases of persistent pulmonary hypertension in the newborn (PPHN) [48,49].

According to a 2012 review of the literature on PPHN, the absolute risk could not be determined, but it was most likely less than 1% and not large enough to warrant stopping the antidepressant drug or reducing its dosage [50].

A different review emphasized the importance of balancing the dangers of untreated mental illness against a potential risk of persistent pulmonary hypertension in the newborn [51].

### 2.3. Serotonin-Norepinephrine Uptake Inhibitors (SNRIs): Venlafaxine and Duloxetine

Even though for SNRI medications there are less data compared with SSRIs, thus far, these medications appear to have similar safety profiles. Venlafaxine use was not associated with an increased risk of major congenital abnormalities, according to a systemic review on the risk of congenital malformations following in utero exposure to either duloxetine or venlafaxine in the first trimester [52]. Although there are significantly less data on duloxetine use, they do not suggest a clinically important increased risk [53,54].

In general, it appears that using duloxetine during pregnancy increases the risk of miscarriage but not the risk of major fetal malformations. Exposure to duloxetine in late pregnancy may be linked to neonatal poor adaptation syndrome, but the extent of this risk is unknown [55].

The use of SNRIs after 20 weeks of gestationis was significantly associated with gestational hypertension. Hence, women on these medications should be monitored for hypertension [56]. However, an increased risk of gestational hypertension has been demonstrated with higher daily maternal doses of venlafaxine.

### 2.4. Norepinephrine Reuptake Inhibitor (NRI, NERI), Noradrenaline Reuptake Inhibitor or Adrenergic Reuptake Inhibitor (ARI): Reboxetine 

Reboxetine use during pregnancy has only been reported in one study (although in combination with other antidepressants) [57].

The author’s findings indicate that there is no evidence linking the usage of SNRIs or NRIs (venlataxine, mianserin, reboxetine, mirtazapine and their combinations) to an increased risk of congenital defects.

Since reboxetine has 15 exposures registered in the literature, it is not possible to make any inferences concerning the teratogenic potential of this drug. 

### 2.5. Serotonin Antagonist and Reutake Inhibitors (SARIs): Trazodone and Nefazodone

Because of its sedative properties, trazodone has been prescribed as a hypnotic, often in conjunction with other antidepressants. It should not be used as a first-line treatment because of scarce reassuring data. In reports from a teratology information service on the offspring of 75 patients with first-trimester exposure to trazodone, there was no increase in malformations compared with unexposed controls [58,59]. In a multicenter teratology information services study, there was no difference in the rates of malformations, miscarriages or stillbirths between offspring exposed to trazodone monotherapy or polytherapy and a group exposed to citalopram, escitalopram or sertraline [60].

Nefazodone is a structural analog of trazodone, and there have only been three studies [59,61,62] conducted on its use in pregnancy, but the reliability of the outcome information is unknown.

### 2.6. Noradrenergic and Specific Serotonergic Antidepressants (NaSSAs): Mirtazapine, Mianserin, Esmirtazapine, Aptazapine and Setiptiline or Teciptiline

Mirtazapine exposure during pregnancy is not expected to increase the risk of congenital anomalies, but the data are considered limited, and consequently, this medication cannot be recommended as a first-line treatment for mood or anxiety disorders. Mirtazapine was not linked to congenital malformations, miscarriage, stillbirth or neonatal death in a nationwide medical birth registry study [63]. In another report, there were no differences in congenital malformations, miscarriage, stillbirth, preterm birth or gestational age or weight between 292 pregnancies with first-trimester exposure to mirtazapine and two control groups [64].

Mianserin has not been systematically studied for pregnancy effects. The use of mianserin as an antidepressant was reported in 48 pregnancies. There were five voluntary abortions, seven miscarriages, one stillbirth and one baby with a malformation [65]. These figures were similar to what was anticipated in the general population.

### 2.7. Norepinephrine-Dopamine Reuptake Inhibitor (NDRI): Bupropion

Although there is evidence to support the use of bupropion during pregnancy, studies on the drug’s impact on cardiac abnormalities have been inconsistent.

A manufacturer registry reported that the prevalence of congenital malformations following first-trimester use of bupropion was 3.6%, not different from the general population rate [66]. The registry’s advisory committee noted the repeated occurrence of heart defects among the collected reports and concluded that the available data were not sufficient. For this reason, the manufacturer conducted a retrospective cohort study and concluded that with the prescription of bupropion in the first trimester, there was no increase in total malformations or cardiovascular defects [67]. In addition, in 2019, a systematic review and meta-analysis concluded that bupropion did not influence the prevalence of congenital malformations, mean birthweight or mean gestational age of exposed babies [68].

### 2.8. Monoamine Oxidase Inhibitors (MAOIs)

#### 2.8.1. Non-Selective MAOI—Hydrazines

Phenelzine

Case reports described pregnancies exposed to phenelzine throughout gestation in which normal offspring were delivered without major malformations [69,70,71]. A review highlighted how in patients taking phenelzine, the consumption of foods containing more than 10 mg of tyramine (aged cheeses and yeast products) could produce a hypertensive crisis [72]. Combinations of phenelzine with another medication can produce a serotonin syndrome [73].

A 2018 review recommended that, to diminish the risk of relapse, phenelzine should be continued if the patient is already pregnant or has not responded to other medications [74].

Isocarboxazid

Isocarboxazid has not been adequately evaluated for pregnancy effects in experimental animals. Human data are inadequate.

#### 2.8.2. Non-Selective MAOI—Non-Hydrazines

Tranylcypromine

Tranylcypromine has been suspected of decreasing uterine blood flow and increasing the risk of adverse pregnancy outcomes. Data to substantiate this suspicion have not been conclusive, since case reports did not establish causation.

The Collaborative Perinatal Project reported increased malformations among 21 mother-child pairs exposed to monoamine oxidase inhibitors during the first trimester [75]. Tranylcypromine was used in 13 of these cases. However, details on the outcomes of the pregnancies involving tranylcypromine were not reported. In two abnormal children born to a woman on tranylcypromine (100–120 mg/day), placental infarcts and congenital anomalies were attributed to a reduction in uteroplacental blood flow [75].

#### 2.8.3. Selective MAO—A

Moclobemide

The data on the use of moclobemide are scarce. There are case reports of normal out-comes after pregnancy exposure [76,77].

#### 2.8.4. Selective MAO—B

Selegine

Selegiline is a selective irreversible inhibitor of type-B monoamine oxidase (MAO-B). This compound has been used as an antidepressant and antiparkinsonian.

Only three case reports are available in the literature [78,79,80] of three women taking selegine during pregnancy. The outcome was positive, with the delivery of three healthy infants and no neonatal complications.

The use of rasagiline was not studied in humans, but in animal studies, it was shown that it increased cardiovascular malformations when administered to pregnant rabbits in combination with carbidopa or levodopa. No malformations occurred in rats [81].

### 2.9. Mood-Stabilizing Medications

Mood stabilizers are a group of drugs with different pharmacological actions. A large number of them are mainly used as antiepileptic medications.

Mood-stabilizing medications play a crucial role in managing bipolar disorder, especially during pregnancy. In particular, these drugs are used to prevent mood fluctuations with up or down peaks. Women diagnosed with bipolar disorder face a considerable risk of experiencing relapses during the postpartum phase.

With a reported pooled prevalence rate of 66%, individuals who do not receive preventative pharmaceutical treatment are especially at risk of relapsing [82,83]. Administering effective pharmacotherapy is thus crucial.

Below is a round-up of drugs that are not antidepressants as such but, in select cases, can be used in the treatment of the depressive phases of bipolar disorder.

#### 2.9.1. Lithium

Lithium, a well-established mood stabilizer, is commonly used as the primary treatment for bipolar disorder [84]. Lithium augmentation therapy has been demonstrated in numerous studies to be beneficial in treating acute episodes of bipolar depression, refractory major depression and delusional depression, as well as reducing relapses of these illnesses. While there are associated hazards, for some women, it is advisable to continue taking lithium during pregnancy, while for others, starting lithium prophylaxis right after birth is the optimal course of action [85]. Using lithium during the first trimester of pregnancy may raise the risk of congenital abnormalities, depending on the dosage. Lithium use in the second and third trimesters was not associated with increased risk [86,87].

Prenatal exposure to lithium raises concerns about cardiovascular malformations, specifically Ebstein anomaly [88]. Although reports suggest a potential increased risk, it remains relatively rare. Studies estimate that while Ebstein anomaly occurs in 1 out of 20,000 live births in the general population, exposure to lithium during the first trimester might increase this risk to a maximum of 1 in 1000 births [89]. Ongoing research into lithium’s reproductive safety involves a substantial retrospective cohort study with 1,325,563 pregnant women, including 663 using lithium during the first trimester [87]. 

The study revealed a modest increase in the risk of cardiac malformations in infants exposed to lithium, with a relative risk of 1.65 compared with unexposed women. Furthermore, a higher risk of right ventricular outflow tract obstruction defects was observed in infants exposed to lithium (compared with unexposed infants). Although the observed relative risk increase appears to be dose-related, causation remains unconfirmed. Despite the relatively low absolute risk, this study reinforces lithium’s teratogenic potential. The advantages of taking lithium during pregnancy, however, might exceed the associated hazards in certain cases. The administration of lithium poses challenges in dosing due to the physiological changes in renal function that occur during pregnancy. Throughout the first and second trimesters, lithium blood levels progressively decline, and in the third trimester, they return to preconception levels [90,91,92].

As a result, in the first and second trimesters, there is a chance that lithium levels will fall below the therapeutic threshold, prompting clinicians to potentially increase the dosage. This adjustment could elevate the risk of lithium intoxication in the subsequent trimester and the postpartum phase. Hence, it is recommended to frequently monitor lithium blood levels during pregnancy and adjust the dosage to maintain levels between the therapeutic range from 0.5 mmol/L to 1.2 mmol/L [93,94]. Prenatal screening, with fetal echocardiography and high-resolution ultrasound around 16–18 weeks of gestation, is recommended for patients on lithium therapy [91].

#### 2.9.2. Lamotrigine

Lamotrigine is often preferred for patients with bipolar II disorder or a history of bipolar traits. It also showed improvement in depressive symptoms in patients with mood disorders. Initial concerns regarding an increased risk of cleft palate or lip deformities in infants exposed to lamotrigine during the first trimester have been largely contradicted by multiple large-scale studies [95].

Analyses comparing pregnancy outcomes and congenital malformation rates among lamotrigine-exposed pregnancies with disease-matched and healthy controls found no significant association with major malformations [96,97]. The latest analysis of the EUROCAT registry in 2016, which significantly increased the study population from 3.8 to 10 million births, found no significantly elevated risk of oral clefts [98]. Researchers in a recent review by Pariente et al. examined a total of 21 studies detailing pregnancy outcomes and rates of congenital malformations. They found that in utero exposure to lamotrigine monotherapy was not linked to an increased risk of major malformations when compared with disease-matched controls and healthy controls.

In comparison to the general population, lamotrigine-exposed pregnancies had similar rates of miscarriages, stillbirths, preterm deliveries and small-for-gestational-age neonates [99]. The results from the latest meta-analyses were consistent with earlier studies; there was no link between lamotrigine monotherapy during pregnancy and a significant rise in risk of major or organ-specific malformations [100]. Overall, according to the available data, lamotrigine appears to be relatively safe as a mood stabilizer during pregnancy.

#### 2.9.3. Valproic Acid

Valproic acid is effective at reducing depressive symptoms in acute bipolar depression. Exposure to valproic acid during the first trimester of pregnancy is associated with significantly augmented risks of major and minor congenital malformations. These include a 20 fold increase in neural tube defects (NTDs), mainly with lumbosacral meningomyelocele, cleft lip or palate, midface hypoplasia, cardiovascular anomalies, genitourinary defects, skeletal and limb malformations, endocrine disorders, developmental delay, growth retardation and microcephaly [91,101]. Additionally, it has been associated with developmental neurocognitive deficiencies, lower IQ, impaired cognition, and increased risks of autism and attention deficit disorders in childhood [102,103,104]. Although valproate is the most effective medication for generalized epilepsies, it should not be used in pregnancy. Combining different drugs may be necessary for alternative treatment.

A recent meta-analysis [105] involving 50,905 pregnancies in individuals with epilepsy compared 1959 women receiving valproate monotherapy to 587 women receiving lamotrigine-levetiracetam duotherapy and 186 women receiving lamotrigine-topiramate duotherapy. Compared with valproate monotherapy, lamotrigine-levetiracetam duotherapy in the first trimster was linked to 60% lower risk of major congenital abnormalities. However, lamotrigine-topiramate duotherapy was not linked to a lower risk.

When treating epilepsy in individuals who may become pregnant, dual therapy with lamotrigine and levetiracetam may be preferable compared with valproate. However, it is unclear if this combination is as effective as valproate. Women of reproductive age are advised against using valproic acid, and if prescribed, comprehensive education on its risks is imperative. Robust contraceptive measures should be employed, and ideally, discontinuation of valproic acid should occur at least six months before planning conception to allow for a safe transition to alternative medications while ensuring mood stability.

#### 2.9.4. Levetiracetam

For the treatment of partial onset seizures, with or without secondary generalization, levetiracetam is authorized as an adjunctive therapy. In addition, in some countries, it is also used as a mood stabilizer, mostly in women of child-bearing age.

The majority of the studies on this agent’s potential effects during pregnancy appear to be encouraging. In a large prospective study based on data from the North American Pregnancy Registry, among 450 children exposed to levetiracetam monotherapy during pregnancy, 11 major congenital malformations were found [106].

A different study based on the epilepsy and pregnancy registries of the United Kingdom and Ireland investigated the safety of levetiracetam exposure in the first trimester, and among the 304 pregnancies exposed to levetiracetam monotherapy, 2 major congenital malformations were detected, and 19 were found when levetiracetam was taken as part of a polytherapy regimen. These findings confirm that levetiracetam monotherapy does not carry an increased risk of malformations [107].

Furthermore, there were no differences observed in the cognitive and language development of children who were exposed to levetiracetam in utero when compared with a control group [108]. According to a recent review by Bromley et al., the prevalence of major malformations for children exposed to levetiracetam (N = 1242), based on the data of 11 cohort studies, was 2.6%, and the prevalence of major malformations in two routine health record studies for children exposed to levetiracetam (N = 248) was 2.8%. Pooled data provided similar risk ratios for women without epilepsy in cohort and routine health record studies. This was sustained by the pooled results from both cohort and routine health record studies when comparisons were made to the offspring of women with untreated epilepsy [108]. In conclusion, levetiracetam appears to be a safe medication during pregnancy since it did not increase the incidence of severe congenital abnormalities in children. It also did not cause any significant neurodevelopmental issues.

#### 2.9.5. Topiramate

It is known that topiramate crosses the human placenta [109]. Topiramate appears to be teratogenic, and it may also accelerate the rate of occurrence of developmental problems. According to the literature, infants exposed to topiramate during pregnancy have a 4.4% risk of developing a major congenital malformation [110]. Exposure to topiramate has an especially strong correlation with a reduced head circumference (18.5%). Microcephaly, low birth weight and cleft palate are among the specific risks it carries (20, 31%).

Topiramate had a dose-dependent risk of oral clefts, with a relative risk for doses ≤100 mg compared with doses >100 mg [111,112]. In a recent review by Ohman et al., a higher rate of malformations was noted with 3.9–4.1% of exposed children having malformations. Compared with children born to women without epilepsy, this was a higher rate. The data showed that children who were exposed to topiramate had a higher risk of developing facial malformations [113]. In summary, topiramate seems to be teratogenic, especially increasing the rate of oral clefts. There are inadequate and contradicting data available on long-term neurodevelopmental outcomes. Therefore, if possible, women of reproductive age should avoid using topiramate as a mood stabilizer.

**Table 1 ijerph-21-00404-t001:** Main characteristics of the drug classes described.

Class	Drug Name	Congenital Side Effects	Side Effects	Is It Safe?
TCAs	Clomipramine, imipramine, amitriptyline, yrimipramine, desipramine, nortriptyline, perfenazine, trazodone	No	Sedation and anticholinergic side effects. (Desipramine & nortriptyline are less anti-cholinergic)	Not first line
SSRIs	Citalopram, escitalopram, fluoxetine, fluvoxamine, paroxetine, sertraline	No	Neonantal side effects	First line
SNRIs	Venlafaxine, duloxetine	No	Gestational hypertension (dose dependent)	Not first line
NRIs	Reboxetine			Scarse data
SARIs	Trazodone, Nnefazodone	No, but scarce data		Not first line, scarce data
NaSSAa	Mirtazapine, mianserin, esmirtazapine, aptazapine, setiptiline or teciptilin	No, but scarce data		Not first line, scarce data
NDRI	Bupropion	Cardiac abnormalities (inconsistent data)		Inconsistent data
Non-selective MAOI: hydrazines	Phenelzine, isocrboxazid		Serotonin syndrome (when phenelzine is combined with other drugs)	Use is justified to diminish relapse (continue therapy if the patient is already pregnant or has not responded to other drugs)
Non-selective MAOI: non-hydrazines	Tranylcypromine	Increased malformations	Decreased uterine blood flow and increase in the risk of adverse pregnancy outcome	Inconsistent data
Selective MAO—A	Moclobemide			Scarce data
Selective MAO—B	Selegine			Scarce data
Mood Stabilizer	Lithium	Cardiovascular malformations (especially Ebstein anomaly) in a dose dependent way		Continue in specific cases monitoring the blood levels
Mood Stabilizer	Lamotrigine	No		Safe to use
Mood Stabilizer	Valproic acid	Neural tube defects and developmental neurocognitive deficiencies		Not Safe
Mood Stabilizer	Levetiracetam	No		Safe to use
Mood Stabilizer	Topiramate	Microcephaly, low birth weight and cleft palate		Not Safe

## 3. Discussion and Conclusions

Maternity care typically places a primary emphasis on pregnancy outcomes, often overlooking the mental well-being of women.

In both the clinical and academic contexts, perinatal depression is acknowledged as a specifier of major depressive disorder, manifesting during pregnancy or within weeks after delivery, and it typically extends to the first year postpartum.

Perinatal depression affects 10–20% of women who give birth worldwide, and risk factors include low socioeconomic status, premenstrual disorders, domestic violence, a history of mental health issues and unintended pregnancies [114].

In the past 10 years, screening tools such as the Edinburgh Postnatal Depression Scale have gained widespread use in clinical settings in order to identify women whose symptoms may require further evaluation for a conclusive diagnosis. Nonetheless, social stigma, cognitive symptoms and subjective reservations about potential custodial issues could deter (expectant) mothers from seeking help or disclosing their symptoms to medical professionals [115].

In the review “Untreated Depression During Pregnancy and Its Effect on Pregnancy Outcomes: A Systematic Review” by Jahan et al., the effect of untreated depression during pregnancy on maternal and neonatal outcomes was examined [7]. Specifically, as demonstrated by Weobong et al., mothers with prenatal depression had a higher risk of peripartum complications including hemorrhaging, complicated vaginal lacerations and placental abnormalities, as well as postpartum complications such as fever, urinary and fecal incontinence and, notably, difficulties initiating breastfeeding [116]. According to Al Rawahi et al., there is a strong link between prenatal depression and cesarean delivery, supported by previous research. It has been found that women suffering from prenatal depression have higher levels of anxiety and fear of labor, resulting in a lower pain tolerance, increased need for epidural analgesia and a higher rate of cesarean delivery [117].

Other studies have shown that preeclampsia, preterm rupture of membranes, premature birth, intrauterine fetal death and fetal growth restriction are all significantly higher among mothers with severe depression compared with those who did not suffer from depression. Smith et al. and Fekadu Dadi et al. [118,119] discovered that depression stimulates the hypothalamic-pituitary-adrenal (HPA) axis during pregnancy, resulting in increased cortisol levels. Chronic stress can also impair the body’s ability to regulate the synthesis of inflammatory proteins. Consequently, inflammation and cortisol are not properly regulated, which can lead to contractions and premature labor.

Experts believe that fetal growth restriction is linked to depression because it alters the neuroendocrine balance. The hormonal end products of hypo- or hyperactivity of the hypothalamic-pituitary-adrenal (HPA) axis (i.e., cortisol and noradrenaline) can influence blood flow in the uterine artery, labor and fetal development and growth [118,119].

Maternal mortality resulting from pregnancy complications has notably decreased in recent decades, thanks to advancements in diagnostic and therapeutic techniques. Nevertheless, as evidenced by multiple studies, mortality linked to perinatal depression is on the rise, emphasizing the necessity for a comprehensive approach that equally addresses the physical and mental well-being of mothers and newborns. In nations where routine maternity care services are available, it is critical to put into practice effective prevention and intervention strategies, despite challenges like stigma and limitation to healthcare access. It may be possible to lessen the negative effects of prenatal depression on health with increased awareness, early diagnosis, well-defined care pathways, clear clinical guidelines, improved diagnostic tools and effective treatment. Nevertheless, achieving this goal demands an interdisciplinary and cooperative approach [120,121,122].

It is essential to identify and address mental health issues before pregnancy, if possible, and subsequently assess and understand the risks, benefits, alternatives and appropriateness of any psychopharmacological treatment, considering the potential consequences of not treating them. If treatment is necessary, then it should not be discontinued, and medications with a known and favorable reproductive safety profile should be selected.

For patients already under treatment with teratogenic or contraindicated drugs during pregnancy, treatment regimens should be adjusted in the periconceptional phase to achieve a stable and euthymic state with the new regimen before conception.

Each patient should receive a personalized treatment, choosing the most appropriate medications for her mental state and minimizing, if possible, the number of drug exposures to the fetus during pregnancy [123].

While selective serotonin reuptake inhibitors (SSRIs), coupled with psychotherapy, continue to be the first-line treatment for perinatal depression, there is a growing focus on new drugs [124].

It is crucial to discuss the dangers, advantages, benefits, alternatives, and appropriateness of psychotropic medications, including the risks of no treatment.

Morbidity and mortality associated with mental health disorders before and during pregnancy can be significantlsy reduced by early screening, diagnosis and intervention.

## Data Availability

Data are contained within the article.

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
