# Peer review of "Depression Treatment in Pregnancy: Is It Safe, or Is It Not?"

_ijerph, 2024, doi:10.3390/ijerph21040404_

Round 1
Reviewer 1 Report
Comments and Suggestions for Authors
In the general analysis of this study, it seems that the author by providing documentation based on past studies tried to direct the reader's mind to the answer to the question raised in the title about the safety of depression treatment during pregnancy.
In a part of the article (lines 160-170), the author has mentioned a good challenge, what are the side effects of depression if it is not treated? But this part, which could be the main weight of this article, has not been discussed much.
I recommend that for the article to be innovative in this discussion, it is necessary to present studies on the risks of untreated depression during pregnancy for the mother and the fetus in the discussion. let the reader's mind be free to reach the answer to the question raised in the title of the article.
Author Response
The consequences of untreated depression were described in the discussion, particularly from point 455 to 478, with the relevant bibliography added (references 124 to 128).
Reviewer 2 Report
Comments and Suggestions for Authors
This review article addresses an important subject regarding the safety of using antidepressants during pregnancy
The manuscript is will written however, the below comments are important
· The study lacks any table of figure to summarize the findings especially for the side effects or the guidelines recommendations or for the sample size of the studies, etc..
· The study lacks information, for example, the authors mentioned in escitalopram in the SSRIs heading, however, nothing was mentioned about this specific drug
· The authors should demonstrate how does this review stand out
· Perhaps a weakness is that it is not a systematic review or a meta-analysis
Author Response
As requested, missing information on drugs in the SSRI category has been added in the relative section (from point 142 to 146, 152 to 155, 159 to 161) and the relevant bibliography has been added (30, 31, 31, 34).
The table (Table 1) with the summary of the characteristics of the various drug classes was added.
Reviewer 3 Report
Comments and Suggestions for Authors
I read with interest the paper provided. I have a concern about the organization of the results. In abstract and introduction you mainly focus on the antidepressants group (which I believed until the results that was ATC N06A - Antidepressants. However, when the results come, the organization started with antidepressants group but soon evolve to a lot of anti-epileptic drugs stated as mood stabilizing medication.
You should rewrite your paper in a broader sense, or remove drugs that are not antidepressants from their definition.
2 - Somewhere in the paper you should justify the methodology for this review. How was made? reading articles and adding information in a paper? Seems that no strict methodology was used. This should be stated and discussed as a limitation - methods as scoping review, systematic or narrative review should be chosen and cleary justified.
References should be carefully confirmed. Adittionaly, repetitions should be avoided.
line 345 - reference is missing (in a recent review)
line 397 - reference is missing. (a recent review)
line 415 - reference is missing (in a recent review)
Author Response
Missing references have been confirmed.
This article is a narrative literature review offering a critical overview of the subject matter and was written by reading articles and adding information.
As already stated in the article, antiepileptic drugs were mentioned not because they are antidepressants as such, but because in selected cases they can be used in the treatment of the depressive phases of bipolar disorder.
Round 2
Reviewer 1 Report
Comments and Suggestions for Authors
No new comments
Reviewer 2 Report
Comments and Suggestions for Authors
Thank you for this nice work